# An Exploration of the Multiplicative Effect of “Other People” and Other Environmental Effects on Violence in the Night-Time Environment

**DOI:** 10.3390/ijerph192416963

**Published:** 2022-12-16

**Authors:** Simon C. Moore, Thomas E. Woolley, James White

**Affiliations:** 1Violence Research Group, Security, Crime & Intelligence Innovation Institute, SPARK, Maindy Road, Cardiff CF24 4HQ, UK; 2School of Mathematics, Abacws, Senghennydd Road, Cathays, Cardiff CF24 4AG, UK; 3Centre for Trials Research, DECIPHer, School of Medicine, Neuadd Meirionnydd, Cardiff University, Heath Park, Cardiff CF14 4XY, UK

**Keywords:** violence, night-time environment, queues, events, alcohol, stress

## Abstract

Background: The characteristics of night-time environments (NTEs) in which alcohol is consumed and that contribute to violence are poorly described. We explore competing explanations for violence in the NTE, with a particular focus on the number of patrons and its association with assault-related visits to a hospital emergency department. Other environmental features including the weather and notable events were also considered. The primary aim was to stimulate debate around the causal mechanisms responsible for violence. Methods: Assault-related ED visits occurring between 8 pm and 4 am were recorded at the University Hospital of Wales, the single Emergency Department (ED) serving Cardiff, Wales, United Kingdom. Footfall was derived from the total number of unique MAC addresses recorded per hour collected from ten wireless fidelity monitoring tools located in the city centre. A narrative review of the literature concerning alcohol and violence informed exploratory analyses into the association between night-time footfall, sporting events, the weather, and other potential predictors of assault-related visits to the ED. We developed analytic methods from formal accounts of queueing. Results: International rugby matches at home, the weather (temperature), national holidays, the day of the week, and number of patrons in the NTE predicted assault-related injury (R^2^ = 0.70), with footfall yielding a positive non-linear exponential association consistent with predictions derived from mathematical models of queueing. Discussion: Assault-related visits to the ED have a non-linear association with the number of people socialising in the night-time environment and are further influenced by the weather and notable events. Opportunities for further research that might inform policy and interventions aimed at better managing NTEs are discussed.

## 1. Introduction

Across a population, the likelihood of violence varies systematically. A number of factors interact at the individual level to increase the likelihood of violence, including alcohol and drug use and poor mental health [1,2,3,4]. Violence, however, is a broad term. It includes violence in response to provocation (reactive violence), or violence that is premeditated and typically used to achieve a predetermined goal (proactive violence) [2]. The context, or environment, defined by the World Health Organisation (WHO) [5,6] as the “physical, chemical and biological hazards that directly affect health and also increase unhealthy behaviours,” will also influence the likelihood of reactive violence, most notably night-time environments (NTEs) in which alcohol is sold and consumed [7,8]. The aim of the current paper was to explore the influence of environmental characteristics on levels of violence. We focused on the weather, significant sporting events, and the number of people present in the NTE.

### 1.1. Enticements in the Night-Time Environment

The NTE is characterised by the consumption of alcohol and other psychoactive substances that are implicated in violence [9,10]. This sets the NTE apart from other times and places where individuals congregate, such as travelling to work or engaging in other commercial activities. However, the availability and consumption of alcohol is not the sole distinguishing feature of NTEs.

NTEs, in which there is a high density of licensed premises, are destinations. They will attract patrons beyond the simple sum of each venue’s attractiveness. Studies in tourism, for example, suggest that entertainment venues compete against one another for trade, but also collaborate to market their locale as a destination to attract out-of-area tourists. Marketing can occur both formally, through paid advertising, and informally through social media and word of mouth [11]. Prospective patrons will assess a destination according to several factors [12], such as the availability of food venues, nightclubs, and bars. The popularity of a destination might also be associated with the likelihood of sexual encounters, which in turn increase the attractiveness of the NTEs [13,14]. This broadens the definition of the NTE beyond a location that is dominated by licensed premises to one offering a range of services and opportunities to socialise [13], collectively referred to as the “business of pleasure” [15]. 

In marketing licensed premises, bars will target young people with lascivious images emphasising drunkenness [14,16]. While the nature, or atmosphere, of the NTE and how it influences alcohol consumption has been considered [14,16], there is likely an additional effect of formal and informal marketing in shaping the population willing to patronise the NTE: they identify with and are attracted to others interested in similar pursuits, otherwise known as homophily [17]. Homophily is the “principle that a contact between similar people occurs at a higher rate than among dissimilar people” and, therefore, that the “cultural, behavioural, genetic, or material information that flows through networks will tend to be localized” [18]. In respect of the NTE, we assume that patrons of the NTE are a relatively homogenous group in respect of demographics, alcohol consumption and reasons for frequenting the NTE. It is an assumption that partly explains the increased prevalence of violence in and around licensed premises: the NTE attracts large numbers of younger people with similar intentions and, by implication, a greater willingness to engage in risk-taking and, therefore, violence.

From a deterministic perspective, if we assume an ensemble of traits, characteristics and circumstances that predispose an individual to be violent, and if the physical environment remains constant, then, as the number of individuals in the NTE increases, the proportion predisposed to violence similarly increases in a linear fashion. This is, however, mediated by the mix of individuals in the NTE. For example, certain cultural or sporting events might attract a particular demographic with traits that predispose them to violence [19,20]. These traits variously involve, but are not limited to, impulsivity, alcohol consumption and a younger age, each of which is associated with aggression [21,22,23]. It is, therefore, feasible that the availability of alcohol together with the way in which an NTE is marketed attracts people with traits that predispose them towards violence. 

### 1.2. Other People

It is reasonable to hypothesize that the on-site consumption of alcohol together with the opportunity to socialise motivate some to patronise premises in an NTE. Assuming this is the case, as the number of operating licensed premises falls, the motivation to remain within the NTE would be expected to decrease, and, therefore, the number of people in the NTE should also fall. If there is a relationship between the number of patrons in the NTE and violence, then we would expect that closing premises will also reduce the number of people in the NTE and that, therefore, levels of violence will also fall. NTEs in which premises’ business hours are restricted show a reduction in levels of violence [24,25], which is attributable to patrons leaving the NTE [26], indicating that the prevalence of violence is a function of the number of people in the NTE. This supports the deterministic account of violence, in that as far as the number of people present increases, so does the number of people predisposed to violence. It does not, however, provide an indication of the nature of events that instigate violence.

The NTE can be crowded and noisy, and exposure to these may elicit stress and, therefore, aggression in people who are at an increased risk of violence [27,28]. One reason for increased violence in NTEs is due to this interaction of alcohol intoxication and crowding, where intoxication undermines usual affiliative behaviours, increases goal blocking and, therefore, elicits an adrenocortical stress response, critical in the aetiology of reactive violence [27,29]. Other factors contributing to elevated stress include observing aversive interactions, including others’ aggression, undesired or uncontrollable events [30], competition when resources are scarce [31,32] and overcrowding [28,33,34]. 

Models of pedestrian behaviour have considerable value, as they can inform the design of environments to offset hazards at times and places when large numbers of people congregate [35]. Such models typically predict emergent pedestrian characteristics [36], including lane formation [37,38]. Most pedestrian models assume identical agents can navigate their environment and avoid collisions. However, a staggering gait accompanies acute alcohol intoxication and is likely to compromise the usually orderly behaviour that emerges as crowd density increases. As entropy increases (predictability decreases and disorder increases), pedestrians would be subject to more stressors, including unwanted collisions and reduced flow [35]. 

One response to resource competition in a disorganised environment is the formation of a queue. Queueing has received considerable attention in operational research [39], in which formal modelling aims to inform the best use of resources to meet demand. However, queues are also social phenomena: those participating in a queue will have expectations on how others should behave, such as not skipping to the front, or reneging, at the expense of others [40]. Subordinates in a queue attain their status from when they join the queue; as those ahead of them are served, their status increases [41]. A queue is, therefore, a social system with a set of shared beliefs concerning the fair distribution of resources according to order of arrival. A notable violation is skipping the queue order, or reneging [40]. When such violations occur, those who queue in an orderly fashion will seek to defend the integrity of the queue, with the most vociferous complaints from those who are closest to the point of intrusion [42], although those ahead of the intrusion may also react to the injustice as well [43]. Furthermore, waiting in a queue elicits stress in those who are queueing [44], and in proportion to the perceived time that they are waiting [45]. This marks a distinction between deterministic accounts of violence and the environmental features in which violence emerges. Violence is partly determined by other people: waiting in line elevates stress and queue jumping represents a violation of social norms and the unfair distribution of resources. Generalising to the NTE, patrons will queue to attain various resources, at fast food outlets, at the bar for alcohol, or for a taxi to get home, points associated with increased violence [8]. However, NTEs differ from formal settings in that they often lack clear queue management, and, therefore, contribute to an uncertain sequence of who should be served and in what order. Our contention is that features of NTE queues may partly explain the prevalence of violence in the NTE.

### 1.3. Queueing

In respect of the NTE, we assume that the rate at which alcoholic beverages (or other resources, such as taxis or fast food) are consumed is uniform and that once a beverage is consumed, the patron will re-join a queue to order a fresh beverage. Thus, the number of individuals within a premise will be in consequence of a random process and in direct proportion with the inter-arrival rate at the bar: the more people present, the lower the inter-arrival time. It is feasible that other probability distributions might better describe processes in the NTE context, but for expository ease, we assume this is sufficient for the purpose intended. Furthermore, in the context of NTEs, it is in bar owners’ economic interest to understaff bars to ensure expenditure on staff salaries is lower than the income gained from sales. This strategy, we will argue, is likely to reduce access to servers and increase competition between those in the queue and, therefore, wait times. 

Consider a multi-server system [39], where patrons form a single queue and are served sequentially. Such a setup is defined according to three factors:

μ, mean service rate and equal to the reciprocal of the expected service time, 1/E[Service-Time],

λ, mean rate of arrival and equal to the reciprocal of the expected inter-arrival time, 1/E[Inter-arrival-Time]

s, the number of servers

From defining these three variables, the probability that the system is empty, $P_{0}$, is
(1)$$P_{0}=\frac{1}{\sum_{n=0}^{s-1}\frac{1}{n!}\left(\frac{\lambda }{\mu }\right)+\left(\frac{1}{s!}\right){\left(\frac{\lambda }{\mu }\right)}^{s}\frac{s\mu }{s\mu -\lambda }}.$$

The average number of customers in the system, $L_{s}$, is
(2)Ls=λμ(λμ)s(s−1)!(sμ−λ)2P0+λμ.

The average amount of time a customer will spend in the system, $W_{s}$, is
(3)Ws=μ(λμ)s(s−1)!(sμ−λ)2P0+1μ.

The average number of customers in the system, $L_{q}$, is
(4)$$L_{q}=L_{s}-\frac{\lambda }{\mu }.$$

The average time a customer spends waiting in the system, $W_{q}$, is
(5)$$W_{q}=W_{s}-\frac{1}{\mu }.$$

Holding μ constant, Figure 1 and Figure 2 demonstrate the effect of increasing the number of servers from two (solid line) to three (dashed line). 

As the inter-arrival time (IAT) decreases, the estimated average wait time in the queue increases. Similarly, as the IAT decreases, the probability that a server is idle (not serving a customer) decreases. Increasing the number of servers increases the server idle probability but has a marked effect on the time customers spend queueing for service.

From the perspective of a licensed premise, a taxi rank or fast-food outlet, there will be a cost in employing staff who remain idle. They are being paid a salary but are not selling the goods, or services, from which profits are made. Queues are good for business. Even as the inter-arrival times of customers fall, and premises are busier, there will be an increase in idle time as the number of servers increases. This provides a motivation for premises in the NTE to keep servers to a minimum to maximise profits. From the perspective of patrons of the NTE, however, wait time in a queue increases exponentially as inter-arrival time decreases (see Equations (1)–(3)), and therefore, we argue, levels of stress and the likelihood of queue jumping will increase. We are not aware of research that has considered the likelihood of queue intrusions as queue length increases. However, as the inter-arrival time of customers decreases with a fixed number of servers, the opportunity for servers to inadvertently service customers in the incorrect order may also increase, as they find it increasingly difficult to keep track of the correct serving order when the bar is replenished with more patrons. 

### 1.4. Summary

From the above discussion, we sought to explore three aspects of "other people" and the environment and their contribution to levels of violence. A deterministic perspective on violence in the NTE would presume the proportion of individuals predisposed to violence would increase as a linear function of the number of people present, and that, therefore, levels of violence will increase as footfall increases. This deterministic account has a notable limitation. It assumes that the individual characteristics promoting violence are simply additive. It is feasible that various events bring different demographics into the NTE, and, therefore, the proportion of individuals predisposed to violence will vary by event type. For example, large sporting events will draw in people who are younger, more impulsive and have a greater interest in alcohol use compared to a regular evening in the NTE where people who frequent licensed premises mix with others attending theatres, cinemas and restaurants [46]. Considering the interpersonal characteristics that might give rise to violence, crowding, and in particular defensive responses that protect the integrity of queues, may also help to explain the relationship between the number of people in the NTE and levels of violence. We explored a formal model of queueing which suggests a non-linear relationship with stress and, therefore, violence. Our aim was to explore the relationship between footfall, sporting and other events, including the weather, in determining assault-related ED visits (ARI).

## 2. Methods

There are, broadly, three ways to capture data on violence: routine police recorded violence, self-reported responses to surveys, and routine ED health data. Self-reported violence data is problematic—the surveys may miss entire sections of the population, such as young people, methods may change and not be comparable over time, and there may be difficulties differentiating between more and less serious incidents [47]. For routine police data, events can only be ascertained if recorded and, therefore, rates will vary according to the police resources available and willingness of victims to report violence to the police [48]. For violence resulting in significant injury, victims will often transfer directly to emergency healthcare services and are lost to police ascertainment [49]. While ED data on assault-related injury will be specific to serious injury, and do not include minor and immediately fatal injuries, they represent a form of routine data that is not subject to the same biases as other methods of assessment and directly measures the health consequences of violence. 

### 2.1. Data

Data were collated and combined from the following sources covering the period: 20 July 2016 to 28 August 2019. 

#### 2.1.1. Assault-Related Injury

The city of Cardiff is served by the University Hospital of Wales and a single ED that is part of a surveillance network making available anonymised assault-related injury data for the purpose of challenging the causes of violence [50,51]. These data include the time, location and circumstances of every patient visiting with an assault-related injury. These data do not contain identifiers, such as patient name, address, or other details. These data were accessed under a data sharing agreement with the Cardiff and Vale University Health Board, and their use is exempt from ethical review and patient consent. 

#### 2.1.2. Estimating Footfall

Wireless fidelity (Wi-Fi) monitoring tools can detect and register probe requests and their unique media access control (MAC) address. In a short period of time, it is possible to count the number of unique MAC addresses in an area and estimate the number of mobile telephones and the number of unique MAC addresses, which, therefore, act as a proxy to the number of people in each location. 

Mobile devices have the facility to connect to a wireless local area network (WLAN). The process of forming a connection first involves a beacon frame; the access point periodically (usually several times each second) broadcasts frames that advertise the MAC address of the access point, the service set identifier (SSID; normally the network name) of the Wi-Fi network, and information on any data rates or encryption that the network imposes. Wi-Fi clients, such as mobile phones, passively pick up these beacon frames and initiate a connection. A second method that is sometimes used involves a probe request frame. Probe request frames are sent by Wi-Fi clients, such as those on mobile telephones, as part of an active scan. A probe request frame can be directed at all Wi-Fi networks available, ostensibly seeking information on what WLANs are available, or it can be directed at a specific network SSID. The latter is usually specific to a previously known WLAN that may hide its SSID. In both cases, the access point will respond with a probe response frame that contains information like that in a beacon frame [52].

The method used to estimate footfall involves only recording probe requests sent by clients and does not record any personal information or any information unique to the mobile phone’s user. Further, as the method is passive, it does not respond to probe requests and does not, therefore, interfere with client–beacon communications. 

Different devices, including printers, cars and computers, can also be Wi-Fi enabled, obscuring the estimated number of people. Other factors may also impact data quality, including sensor range and (rarely) MAC address collisions. To account for this in the available footfall data, probe requests from the same device across consecutive five-minute periods are counted only once. This means non-mobile devices are counted only once. Furthermore, there is a limit of four on the number of times a probe request can appear in a day. For example, if a device is detected in a single location for a five-minute period at six different points in time in a day, it is only counted four times. Further details, including data availability, are available from the Consumer Data Research Centre [53].

#### 2.1.3. Estimating Footfall in Cardiff

There are 15 Wi-Fi sensors in Cardiff. One sensor was dropped as it was in an out-of-town retail park approximately 3.7 miles outside of the NTE. Another sensor had one single hour of data and was also dropped. Some sensors are public and always-on; others are associated with commercial properties, and data availability is accordingly intermittent and follows commercial operating hours. The theoretical maximum number of hourly observations from a single sensor was 27,072 over the study period. The percentage of the maximum observations averaged across all 13 remaining sensors was 69.39%. Three sensors were intermittent, with only 18.88% data available, and were dropped. For the remaining ten sensors, data availability was 92.69%, and the number of sensors operational by hour was constant (mean = 9.17 sensors, SD = 0.005, Min = 9.17, Max = 9.19), suggesting hour of operation was not systematically associated with time of day. The data were supplied by hour, with the estimated total number of unique MAC addresses for each sensor and each hour. To derive a proxy footfall index, we averaged the number of unique MAC addresses that connected across the available and operational sensors in Cardiff for each hour (Figure 3) and for the per session (8 p.m. to 4 a.m.) analysis, we summed across each sensor. The mean is reported in Table 1. 

#### 2.1.4. Sporting Events

To explore whether sporting events influence violence, the dates of international rugby sporting events at home and away were collected. The Welsh national team plays other international rugby teams in the Cardiff stadium, with a capacity of 75,500, which is typically sold out. The other games will be played away at the other team’s stadium. Given Cardiff has a local total population of approximately 470,000, these events attract a significant number of people into the city who are interested in rugby. Sporting events are typically associated with alcohol consumption [54], especially international rugby events [55].

#### 2.1.5. Weather

Temperature is associated with violence [45], and we, therefore, explored this in models. Daily weather data were scraped from the National Oceanic and Atmospheric Administration (NOAA), where data are collated from over 9000 weather stations worldwide [56]. Daily data on rainfall and temperature from the Cardiff Weather Centre (ID 37170) were accessed.

#### 2.1.6. Public Holidays

In England and Wales, there are typically eight public holidays, usually on a Monday. Previous research suggests that these bank holidays (so called as banking institutions are usually closed) are used as a recovery day for those who consume alcohol, implying the day before a bank holiday is one that is characterised by excessive alcohol consumption [46]. 

### 2.2. Analytic Strategy

Table 1 presents the descriptive statistics. All data were restricted to evenings (8 pm to 4 am), the usual busy operating times for premises licensed for the sale and onsite consumption of alcohol, and the footfall index was summed for each of the 1129 eight-hour sessions, as were ARIs from ED data. Weather was averaged across each session. We further included binary indicators for bank holidays and the day before a bank holiday, as well as days when the Welsh rugby team played at home or away. The relationship between footfall index and ARI was modelled using non-linear least squares regression. We initially explored the relationship between independent variables and footfall using a Prais–Winsten regression, adjusting for serial autocorrelation (Table 2).

Models exploring the association between environmental features and ARI were fitted using least squares. In the first model, a simple linear relationship between footfall and ARI was assessed (Table 3, I) and then compared to a non-linear exponential relationship between footfall and ARI. Referring to Equation (6), $y_{i}$ is a vector of the observed assault totals, $x_{i}$ is the footfall index regressor, and $\epsilon_{i}$ the disturbance term. Subsequent models included covariates for rugby at home and away, weather and bank holidays. Model selection was informed by R^2^, Akaike information criterion (AIC), and the Bayesian information criterion (BIC).
(6)$$y_{i}=\beta_{1}\left(\beta_{2}{}^{x_{i}}\right)+\epsilon_{i}$$

## 3. Results

Figure 3 graphically presents average city centre footfall and average ED ARI across a week and by hour. Notably, peak ARI did not correspond with peak footfall, highlighting that those frequenting the city centre at night, compared to daytime, are likely drawn from different communities. Table 1 presents summary statistics. 

Initially, we applied a generalized least-squares linear regression model to estimate the influence of independent measures used in modelling log footfall. The model errors were serially correlated and assumed to follow a first-order autoregressive process (Table 2). The Prais–Winsten AR(1) regression yielded a significant positive effect when rugby was played at home, but not when the match was played away. Warmer weather reduced the number in the NTE, whereas rainfall increased footfall. 

Table 3 present results from four models. In Model I, a significant linear relationship between footfall and ARI was identified. However, in Model II, the exponential relationship between footfall and ARI performed better, according to all fit indices. Rugby matches at home attract greater numbers of people to the NTE (Table 2) but have an additional effect over and above the effect of footfall on ARI. This is consistent with the assertion that certain events attract individuals with traits that predispose them to violence. Higher temperature increases the likelihood of ARI, but not greater rainfall. The latter might be explained by the prevalence of rain in Cardiff, which is nearly constant. Further, the day before a bank holiday, when greater alcohol use was expected, yielded a greater increase in ARI. 

## 4. Discussion

We sought to explore the relationship between the number of people frequenting a city centre environment at night and levels of ARI recorded in a local ED. The overarching aim was to explore a theoretical perspective that accounted for environmental features, including other people, in the aetiology of violence. Our analyses suggest that the number of people present in the NTE is associated with the number of ARIs in ED—in and of itself, an important observation given that the bulk of research in this area fails to account for footfall. For example, a number of studies consider the effect of interventions to reduce harm in NTE [57], but without providing insights into whether these interventions drive patrons away from the NTE, or reduce harm through changing behaviour. This is a key difference, as it points to different methods of intervening to reduce ARI in the NTE.

We framed the role of alcohol in the NTE both as an attractor and cause of disinhibition—an attractor, as the business of pleasure variously attracts people with a greater proclivity for violence. We further suggested that events within the NTE would further this homophilous concentration of traits and consumption profiles predisposing patrons to violence, an assertion partially supported by our observation that bank holidays, typically used as an excuse to excessively consume alcohol [46], and rugby events at home increased violence, over and above what might be expected from the impact of greater footfall.

The notion that alcohol consumption is implicated in the prevalence of harm in NTEs is poorly operationalised. The assumption has been that greater alcohol use, and, therefore, the mis-selling of alcohol, means greater levels of violence. While there is evidence to suggest that alcohol causes disinhibition and, therefore, increases the propensity to be violent [58,59], the decision to drink excessively is, itself, likely predicated on a behavioural trait of impulsivity [22], a trait more apparent in those who are younger [23]. An alternative explanation is the association between the NTE and violence is reverse cause. Rather than the NTE causing violence, it may be that individuals with a pre-existing propensity for violence choose to congregate en masse in NTEs, increasing the likelihood of violence. We, therefore, argue that homophily, where similar people group together, might mean those attracted to NTEs share similar traits, such as youth or an interest in alcohol consumption, and pursue other opportunities available in the business of pleasure [14]. Attracting similar people to the NTE, and into an environment that encourages boisterous and lascivious behaviour, would appear to play an important role in the aetiology of violence and may further reinforce social norms on the acceptability of violence within the NTE [60].

We demonstrated a non-linear relationship between footfall and violence. We argued that this association is in part due to failures in the mechanisms typically used to distribute resources fairly in queues. While the observation that an increase in NTE footfall is associated with greater ARI is robust, we have assumed that behaviour in queues results in a non-linear association with ARI. We have provided a theoretical rationale to explore this further: that stress due to observing transgressions in fair queueing would be proportional to the number of those in a queue (a proxy of which is footfall), and building on queueing theory, an exponential relationship between footfall and violence. It is feasible that queuing theory is as applicable to the purchase of alcohol as it is to other scenarios where resources might be scarce: taxi ranks and fast-food outlets, locations in which violence is also prevalent in the NTE [8]. It also suggests interventions that can reduce queuing time (e.g., increasing bar staff numbers), reduce queue jumping, or better manage the distribution of resources (e.g., mobile app service) may reduce the risk of violence. Ledbetter [44] reiterates the aversive nature of queues, and provides a taxonomy of interventions that reduce the negative aspects of queuing. These include distraction, comfort, facilitation of interpersonal interaction, communicating wait times to patrons and visually separating inequitable wait queues. The latter may be particularly pertinent in EDs, where clinical need will determine how long patients wait to be seen [61]. Taxi rank marshals, who manage queues for taxis in the NTE, have been similarly recommended as personnel able to intervene and reduce queue jumping and violence in the NTE [62]. These theoretical predictions from queuing theory do, however, match the empirical observations made here.

Queueing theory provided additional insights into potentially conflicting motivations of those who operate licensed premises and those who patronise them. As the probability that a server is idle is influenced by the number of servers present, there is likely a profit motive to understaff rather than overstaff, as far as possible. Conversely, the queue wait time for patrons is strongly influenced by the number of servers present, suggesting a small reduction in the number of servers disproportionately increases wait time. Similar effects would be likely in taxi ranks and fast-food outlets. The UK Licensing Act 2003 imposes duties on license holders to prevent crime and disorder, and to maintain public safety, but there are no provisions on how licensable activity should scale as the number of patrons increases and limit the number of patrons allowed in premises. If further research confirms the observations made here, there will be a need to address the design and operation of services in the NTE.

The analyses presented here were exploratory but offer some robust insights that inform the relationship between NTEs and violence. Notably, the variation in footfall was positively associated with assault-related injury in a non-linear fashion predicted by formal models for queues. There are, however, limitations to the analyses presented. Most significant is the lack of information on the nature of those frequenting the NTE in question. Many of the assumptions and interpretations made here could be confirmed through additional research aimed at providing a better understanding of who, along with how many, patronise the NTE. Further research could also continue the modelling undertaken here to work through scenarios where the integrity of queues is maintained. Models of pedestrian flow [63,64], for example, have value in informing the physical design of the environment to reduce crowding and congestion and, therefore, inform interventions that might reduce harm, and this extends to queueing behaviour [65,66].

The relationship between the environment, the individual, and the emergent phenomena of violence, can be understood from a variety of perspectives. Observational epidemiological “black-box” methods would consider rates of violence across potentially large numbers of individuals with common characteristics, such as age or gender, but do not typically consider the mechanisms, pathogenesis, or role of the environment in the aetiology of violence in the NTE. The approach we adopted here was to explore intermediate behavioural processes that causally link the activity of individuals to emergent measurable violence. It is an approach that has similarities to those found in statistical mechanics [67] and related enquiry [68]. For example, Maxwell, building on Bernoulli’s Hydrodynamica and later generalisations by Boltzmann, described the movement of, at that time, theoretical particles of motion to explain the emergent phenomena of pressure. We can measure the number entering the NTE, and the number of ARIs in the local hospital, and like the kinetic theory of gas, formally explore mechanisms relating input to output. However, there are also corollaries with Durkheim’s concept of anomie [69]. Anomie describes a state characterised by a lack of social cohesion and an absence of ethical or social standards. Without clear expectations on how a person should behave, according to Durkheim, moral confusion promotes deviant behaviour. It is a concept that is aligned to our use of entropy to explore how the breakdown of orderly queueing behaviour within a crowed NTE may promote violence, and how the deviant behaviour of queue jumping weakens the social cohesion of patrons within the queue.

## 5. Conclusions

Violence in night-time environments might be attributable to a failure to abide by the social requirements of queueing. The current exploratory study suggests meaningful lines of enquiry, but any subsequent research should seek to go beyond simple correlations and formally consider the interaction of individual characteristics and features of the NTE that may precipitate violence. If this relationship is confirmed in additional research, there will be a need to encourage proper queue discipline or better manage queues in the NTE to reduce violence. 

## Figures and Tables

**Figure 1 ijerph-19-16963-f001:**
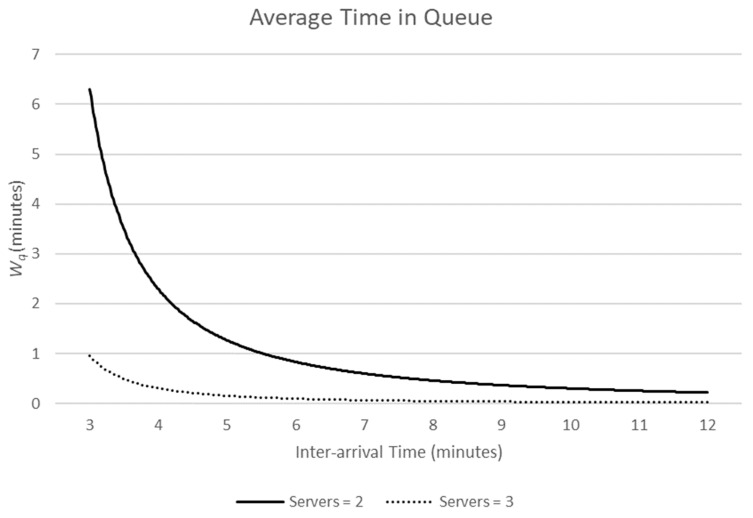
As the rate of customers joining the queue increases, the average wait time increases exponentially, but wait time is reduced as the number of servers increases.

**Figure 2 ijerph-19-16963-f002:**
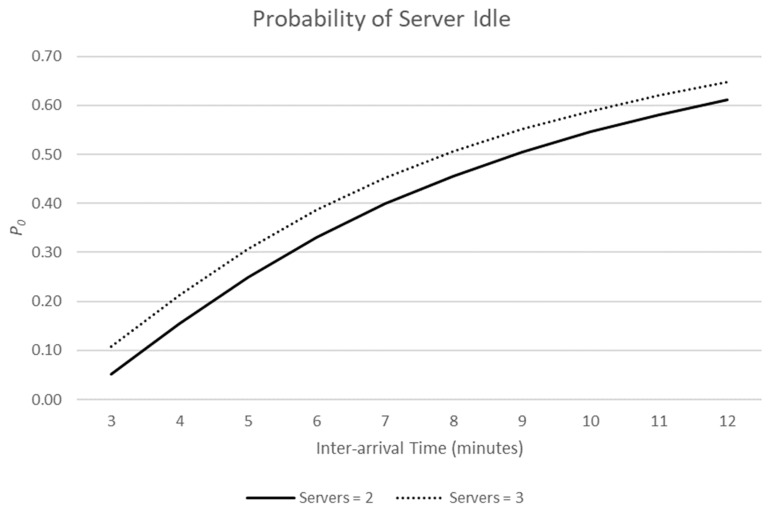
As the number of servers increases, the probability that a server is idle increases.

**Figure 3 ijerph-19-16963-f003:**
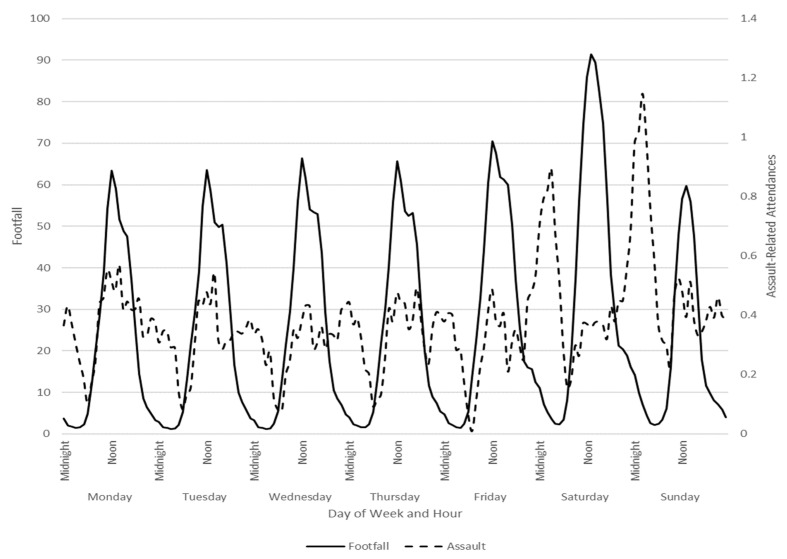
Hourly assault-related injury and footfall index.

**Table 1 ijerph-19-16963-t001:** Descriptive statistics.

			95% CI	
	Mean	Proportion ^1^	Lower	Upper
Assault-related visits	2.210		2.089	2.331
Female	0.503		0.458	0.548
Male	1.707		1.602	1.813
Footfall index (evenings)	514.19		490.33	538.06
Temperature (°F)	52.518		52.005	53.032
Rainfall (inches)	0.088		0.076	0.100
Rugby at home		0.017	0.010	0.026
Rugby away		0.013	0.007	0.022
Bank holiday		0.022	0.014	0.033
N = 1129 8 p.m. to 4 a.m. sessions				

Notes: ^1^ Proportion of all sessions in which the designated event occurred.

**Table 2 ijerph-19-16963-t002:** The association between rugby matches, days of the week, holidays and the weather on city centre footfall.

			95% CI		
Log Footfall		β	Lower	Upper	*p*
Rugby at home		3.704	2.870	4.538	<0.001
Rugby away		0.045	−0.862	0.952	0.923
Temperature		−0.037	−0.064	−0.011	0.006
Rainfall		0.627	0.061	1.194	0.030
Bank holiday		−1.196	−2.008	−0.383	0.004
Bank holiday (day before)		3.304	2.492	4.116	<0.001
Day of week (reference = Sunday)					
	Monday	0.751	0.387	1.114	<0.001
	Tuesday	0.703	0.261	1.145	0.002
	Wednesday	1.232	0.758	1.706	<0.001
	Thursday	1.480	1.008	1.951	<0.001
	Friday	6.575	6.132	7.018	<0.001
	Saturday	7.741	7.366	8.116	<0.001
Constant		5.351	3.903	6.800	<0.001
N = 1129 8 p.m. to 4 a.m. sessions					
Prais–Winsten AR(1) regression (iterated estimates)					
Durbin–Watson statistic (original)		0.900			
Durbin–Watson statistic (transformed)		2.050			

**Table 3 ijerph-19-16963-t003:** Association between footfall and ARI.

				95% CI					
Model			β	Lower	Upper	*p*	R^2^	AIC	BIC
I	Footfall	β_1_	0.0034	0.0033	0.0036	<0.001	0.5576	4788.409	4793.439
II	Footfall	β_1_	1.4419	1.3243	1.5595	<0.001	0.6151	4633.211	4643.270
		β_2_	1.0007	1.0007	1.0008	<0.001			
III	Footfall	β_1_	0.8109	0.3865	1.2352	<0.001	0.6346	4586.422	4626.655
		β_2_	1.0009	1.0007	1.0011	<0.001			
	Rugby at home		2.5420	1.6283	3.4556	<0.001			
	Rugby away		1.3955	0.4501	2.3409	<0.01			
	Temperature (°F)		0.0141	0.0047	0.0234	<0.01			
	Rainfall (inches)		−0.0806	−0.6229	0.4616	0.770			
	Bank holiday		0.9236	0.1883	1.6589	0.014			
	Bank holiday (day before)		0.8678	0.1336	1.6020	0.021			
IV	Footfall	β_1_	0.0608	−0.0920	0.2136	0.435	0.6984	4383.933	4459.369
		β_2_	1.0016	1.0005	1.0027	<0.001			
	Rugby at home		2.0416	1.1833	2.8999	<0.001			
	Rugby away		0.5812	−0.2898	1.4521	0.191			
	Temperature (°F)		0.0186	0.0069	0.0302	<0.01			
	Rainfall (inches)		−0.0518	−0.5476	0.4440	0.838			
	Bank holiday		1.0927	0.4008	1.7847	<0.01			
	Bank holiday (day before)		1.4343	0.7407	2.1280	<0.001			
	Sunday		0.3768	−0.3711	1.1248	0.323			
	Monday		0.3230	−0.4349	1.0809	0.403			
	Tuesday		0.2701	−0.4823	1.0224	0.481			
	Wednesday		0.3565	−0.4042	1.1172	0.358			
	Thursday		0.4196	−0.3380	1.1773	0.277			
	Friday		2.0294	1.1904	2.8684	<0.001			
	Saturday		2.8472	1.9644	3.7299	<0.001			

## Data Availability

The Emergency Department data used in this study is available through application to Digital Health and Care Wales (https://dhcw.nhs.wales). The mobile phone data is available through application to the Consumer Data Research Centre (https://www.cdrc.ac.uk).

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
