# Peer review of "An Exploration of the Multiplicative Effect of “Other People” and Other Environmental Effects on Violence in the Night-Time Environment"

_ijerph, 2022, doi:10.3390/ijerph192416963_

Round 1

Author Response

MANUSCRIPT ID: ijerph-2029130

MANUSCRIPT TITLE: An Exploration of the Multiplicative Effect of “Other People” on

Crowds, Queues, and Violence in the Night-Time Environment

Very many thanks for the reviewers who took time to scrutinise our manuscript. Their comments and insights are particularly useful in developing our work further.

Reviewer 1

We are very grateful to Reviewer 1 for their incisive and valuable comments. The intention of the paper was exploratory, one to provoke greater consideration of the events withing NTEs that might promote violence. We instead, in places, presented the work as one that sought to test specific hypotheses. Based on these comments we have made considerable revisions, in the hope doing so addresses the valid concerns raised.

Notable revisions include changes to the title,

“An Exploration of the Multiplicative Effect of “Other People” and Other Environmental Effects on Violence in the Night-Time Environment”

We have also revised the abstract.

  1. H2 and H3 are vague. They need to propose particular relationships, not just the existence of relationships. Given that hypotheses suggest that knowledge of an independent variable

condition can predict a dependent variable condition, these specific conditions need to be

identified.

We have revised the hypotheses to emphasise the more exploratory nature of the paper.

  1. With regard to H2, in what ways do homophily and proportion of individuals predisposed

to violence vary by event type?

We have revised the Summary section and outlined several examples. We also cite literature from tourism and alcohol epidemiology that underpins this assumption.

What types of events are predicted to have higher proportions and what types of events are predicted to have lower proportions? Lines 424-427 of the manuscript refer to a “hypothesis partially supported by our observation,” and identifies bank holidays and rugby games as predictors of violence within crowds. But, the hypothesis, itself, does not specify these specific types of events as factors associated with violent behavior.

We have added reference to Moore et al. (2014) providing evidence that alcohol use is increased around sporting events, important in generating violence.

  1. With regard to H3, what is the predicted exponential relationship? Is it a quadratic (x2) curve? Is it a cubic (x3) curve? Is it a logarithmic curve? etc.

We presume an exponential relationship of the form ex.

  1. The study’s conclusion addresses the need to manage queues. But, data presented in the manuscript does not pertain to the queues, themselves; only to footfall. The assumption that footfall manifests itself as the length and orderliness of a queue challenges criterion-related internal validity. Although the likelihood of disorganized queues does likely increase as footfall increases, this is not necessarily the case (e.g. For example, Ledbetter (1) explains that the Disney company has implemented queue activities to keep people content while waiting in long lines at its theme parks.). So, the authors cannot validly use footfall as an indicator of queue disorder.

Very many thanks for the Ledbetter reference. We agree that the analyses we undertook cannot be used to infer a formal relationship between queues and violence, we have realigned the Discussion to account for this more transparently and explicitly referred to internal validity.

I do not mean to imply that these two factors make the manuscript unpublishable. They do, however, need attention. The first can be easily addressed by modifying the hypotheses. The second may require reframing of data or acknowledgement of the threat to internal validity as a limitation of the study.

We hope the more relaxed Summary section in the Introduction better describes the exploratory nature of the paper.

Aside from the major concerns already identified, I noticed the following minor issues that the authors may wish to consider.

  1. The manuscript contains some incomplete and run-on sentences. Run-on sentences appear in Lines 262-265 and 322-324. Incomplete sentence appears in Lines 212-213, 413-414 and 448-451. (For the last of these, simply changing the word “being” to “is” will remedy the problem.)

Again, thank you for such a diligent review. “Being” is replaced with “is” where pertinent.

  1. Changes to sentence structure and wording in various spots throughout the manuscript would

increase its readability.

  1. In Line 84, authors should add the word “which” before “in turn.”
  2. In Lines 192 and 194, authors should replace the word “and” with “which.”
  3. In line 304, “it’s” should be “its.”
  4. The sentence in Lines 401 and 402 should state, “Consistent with expectations, higher temperature, not greater rainfall, increases the likelihood of ARI.”

These changes have been made, the manuscript has been further revised to hopefully increase readability.

  1. I am unsure why the authors use healthcare settings as an example in Line 162. The contexts described in the manuscript have very little, if any, resemblance to healthcare settings. Perhaps the authors can provide a more general example than that of healthcare or find an example related to a setting somewhat similar to the bars that they discuss.

In Emergency Care, patients are triaged and will wait in proportion to clinical need. The issue being that those with low acuity will see other patients being seen by clinical staff before them, and crowding and aggression in such clinical facilities is unfortunately common. It is a point that generalises the possible queue length and aggression argument and we now return to this in the Discussion.

  1. Line 390 should specify home matches, as, according to Table 2, away matches have a very high p value.

“at home” has been added across the manuscript.

  1. The information conveyed in Figure 3 is extremely powerful. I believe, however, that the figure should appear in the Results, not the Methods, section of the manuscript.

This has been moved.

  1. The authors should replace the error messages that appear in Lines 210-211, 333, and 393, with the correct text.

Apologies for this, it has been corrected.

I would also like to highlight two portions of the manuscript in which I believe the authors provided impressive reflections upon their work. I encourage them to expand upon these explanations as suggested below.

  1. In lines 145-146, the authors state, “As entropy increases (predictability decreases and disorder increases) pedestrians would be subject to more stressors.” The very prominent sociological concept of anomie, first acknowledged by theorist Emile Durhkeim (2), describes this relationship between unclear norms and chaos. The authors may wish to incorporate this term into their text.

This is a very useful link to make. There is a lot that could be written here, and we have added a paragraph to the discussion concluding with the idea that anomie might have value in future research:

The relationship between the environment, the individual, and the emergent phenomena of violence, can be understood from a variety of perspectives. Observational epidemiological “black-box” methods would consider rates of violence across potentially large numbers of individuals with common characteristics, such as age or gender, but do not typically consider the mechanisms, pathogenesis or aetiology of violence in the NTE. At best, the environment is a convenience describing an ecology the epidemiological study of which provides little insight into formal causal mechanisms. The approach we have adopted here is to explore intermediate behavioural processes that causally link the activity of individuals to emergent measurable violence. It similarities to an approach found in statistical mechanics (51) and related enquiry (52). For example, Maxwell, building on Bernoulli’s Hydrodynamica and later generalisations by Boltzmann, described the movement of, at that time, theoretical particles motion to explain the emergent phenomena pressure. However, there are also corollaries with Durkheim’s concept of anomie (53). Anomie describes a state characterised by a lack of social cohesion, and ethical or social standards. Without clear expectations on how a person should behave, moral confusion promotes deviant behaviour. It is a concept that is aligned to our use of entropy to explore how the breakdown of orderly queueing behaviour may promote violence, that the deviant behaviour of queue jumping weakens the institution of the queue.

It would seem that in the development of models of behaviour, some of which borrow from particle physics, sociological theory can help shape insights and parameterise models. It is something that would be useful to revisit in subsequent work.

  1. The authors acknowledgement that they cannot determine whether NTE causes propensity toward violence or those with propensities toward violence tend to frequent NTEs is wonderful. Methodologically, the term “causal time order” refers to this lack of clear directionality. Including this term will highlight the authors’ abilities to notice this methodological issue and, thus, their attempts to remain as pragmatic as possible when explaining the value of their results.

We have referred to this in the Conclusion:

Violence in night-time environments might be attributable to a failure to abide by the social requirements of queueing. The current exploratory study suggests meaningful lines of enquiry, but any subsequent research should seek to go beyond simple correlations and formally consider the causal time order. If this relationship is confirmed in additional research, there will be a need to encourage proper queue discipline or better manage queues in the night-time environment to reduce violence.

REFERENCES

  1. Ledbetter, Jonathan L., et al. “Your Wait Time from This Point Will Be . . .” Ergonomics in

Design: The Quarterly of Human Factors Applications, vol. 21, no. 2, 2013, pp. 22–28.

  1. Durkheim, Emile. The Division of Labor in Society. (G. Simpson, Trans.) New York: The

Free Press. 1893/1960.

Reviewer 2 Report

I would just like to congratulate you on your work, since it is very difficult with tail analysis to find models that respond to non-exponential distributions.

The authors' main hypothesis is that stress-related reactive aggression would be proportional to queue length and thus follow an exponential relationship. This prediction is confirmed by using a mathematical queuing model to examine the association between the number of people out at night (total number of MAC addresses of network cards or devices) and emergency service records of assaulted people. These results can be very useful for designing policies and interventions to reduce waiting time and queue-jumping. However, care must be taken because they assume that inter-arrival times are exponential, which implies that they occur randomly (Poisson entry process), and this is not very reasonable when customers have very similar needs. In these cases, it would be important to have other queuing models that use other probability distributions, non-exponential distributions, as is the case. Therefore, the results of this study are very useful for developing a model of these characteristics, a new theoretical approach to the subject.

Author Response

Reviewer 2

I would just like to congratulate you on your work, since it is very difficult with tail analysis to find models that respond to non-exponential distributions.

The authors' main hypothesis is that stress-related reactive aggression would be proportional to queue length and thus follow an exponential relationship. This prediction is confirmed by using a mathematical queuing model to examine the association between the number of people out at night (total number of MAC addresses of network cards or devices) and emergency service records of assaulted people. These results can be very useful for designing policies and interventions to reduce waiting time and queue-jumping. However, care must be taken because they assume that inter-arrival times are exponential, which implies that they occur randomly (Poisson entry process), and this is not very reasonable when customers have very similar needs. In these cases, it would be important to have other queuing models that use other probability distributions, non-exponential distributions, as is the case. Therefore, the results of this study are very useful for developing a model of these characteristics, a new theoretical approach to the subject.

Our use of queueing theory aims to demonstrate the nature of the relationships between interarrival rate, wait time and server number. In it’s application to the NTE, the interarrival rate is assumed to rely on a random process. Which is a reasonable assumption, given that in the NTE there will be many locations that can service patrons. Presumably they would find establishments where the wait time is minimal. However, our estimates could be at the lower bound, as some establishments might be more popular than others, exacerbating the wait time in queues. It is feasible that other models relying on other probability distributions would better capture the nature of queues and queueing in the NTE. We have added to the Queueing section:

“Thus, the number of individuals within a premises will be in consequence of a random process and in direct proportion to the inter-arrival rate at the bar: the more people present the lower the inter-arrival time. It is feasible that other probability distributions might better describe processes in the NTE context, but regard our current approach sufficient for the expository purpose intended.”

Reviewer 3 Report

Thank you for the opportunity to read your article about the multiplicative effect of "other people" in night-time environments.

After reading your article in full a couple of times, I am still having a hard time connecting all the dots with what you set up in your introduction, to what you studied, to what you concluded. There was a lot of emphasis in your intro on queues, alcohol, homophily, and number of people in the environment. I was surprised to then see the data you collected, particularly weather. It would be helpful if temperature is mentioned in the introduction. I think the other data collected made sense with what you discussed in your introduction. Then in your discussion, you brought back up queues, yet this did not seem to be included in any of the ways you measured your data. For example, you did not include information on number of servers or other employees in these places of business nor the length of queues, etc. I agree that more detailed studies would need to be conducted to look at demographics of individuals in the NTEs as well as looking more closely at queueing theory.

I would like to see you rewrite your introduction to flow more with the data you collected in your methods/study design, including temperature. I would then like to see queueing theory brought up less as it relates to your study and more for future directions of studies. I do not think you can draw the conclusion you did from your study.

There were a few points in the article that had Error! Reference source not found. Not sure what these were about, but it would be helpful to clear these up.

Line 418 appears to be an incomplete sentence with "A critical stage in developing complex intervention"

Line 464, do not capitalize "While" in "However, while the UK Licensing Act 2003..."

Author Response

Reviewer 3

Thank you for the opportunity to read your article about the multiplicative effect of "other people" in night-time environments.

After reading your article in full a couple of times, I am still having a hard time connecting all the dots with what you set up in your introduction, to what you studied, to what you concluded. There was a lot of emphasis in your intro on queues, alcohol, homophily, and number of people in the environment. I was surprised to then see the data you collected, particularly weather. It would be helpful if temperature is mentioned in the introduction. I think the other data collected made sense with what you discussed in your introduction. Then in your discussion, you brought back up queues, yet this did not seem to be included in any of the ways you measured your data. For example, you did not include information on number of servers or other employees in these places of business nor the length of queues, etc. I agree that more detailed studies would need to be conducted to look at demographics of individuals in the NTEs as well as looking more closely at queueing theory.

Many thanks for these comments. These overlap with Review 1’s comments and we have responded there.

I would like to see you rewrite your introduction to flow more with the data you collected in your methods/study design, including temperature. I would then like to see queueing theory brought up less as it relates to your study and more for future directions of studies. I do not think you can draw the conclusion you did from your study.

We agree, and the discussion has been revised to highlight limitations.

There were a few points in the article that had Error! Reference source not found. Not sure what these were about, but it would be helpful to clear these up.

These have been correct.

Line 418 appears to be an incomplete sentence with "A critical stage in developing complex intervention"

Thank you, this has been corrected.

Line 464, do not capitalize "While" in "However, while the UK Licensing Act 2003...

This has been corrected.

Round 2

Reviewer 1 Report

The authors' revisions, in my opinion, demonstrate thoughtful consideration of my comments. Their explanation that they intend this exercise to be an exploratory analysis as well at the manuscript's new title helps to "frame" the manuscript differently than it was perceived in its earlier version. I still believe that the the investigation, beginning with the statement of hypotheses, needs further refinement if and when the authors develop it into an inferential study. However, given the current manuscript's purpose, it is strong enough to meet the standards for publication in the IJERPH.

Author Response

Many thanks for these positive comments. We have substantially revised the manuscript and, in addition, alluded to the kinetic theory of gases in the Discussion. We feel there is similarities between historical theorising on the movement of, at that time, unobservable particles and formal models describing the relationship between, for example, pressure and heat.

Reviewer 3 Report

Thank you for the revision to your previous manuscript. I appreciated the edits that addressed the comments from the reviewers' reviews. Your manuscript flows much better now.

There is some editing needed with the revision before publication. For example, delete the word "However" after "Furthermore" in line 192. Reword lines 254 to 258. The sentence doesn't make as much sense and doesn't read well with the way it is currently worded and structured. Reword the sentence in lines 263 to 265. Do you mean to say "We therefore seek to explore...?"

Author Response

Many thanks, we have proof read the manuscript with care and have improved the presentation.